# Functional Microfiber Nonwoven Fabric with Copper Ion-Immobilized Polymer Brush for Detection and Adsorption of Acetone Gas

**DOI:** 10.3390/s22010091

**Published:** 2021-12-23

**Authors:** Yung-Yoon Kim, Kazuya Uezu

**Affiliations:** 1Graduate School of Environmental Engineering, The University of Kitakyushu, 1-1 Hibikino, Kitakyushu 808-0135, Japan; z8dab002@eng.kitakyu-u.ac.jp; 2Faculty of Environmental Engineering, The University of Kitakyushu, 1-1 Hibikino, Kitakyushu 808-0135, Japan

**Keywords:** radiation-induced graft polymerization, VOCs, acetone, copper ions, 4-picolylamine, coordination bonding, microfiber nonwoven fabric

## Abstract

The detection and removal of volatile organic compounds (VOCs) are emerging as an important problem in modern society. In this study, we attempted to develop a new material capable of detecting or adsorbing VOCs by introducing a new functional group and immobilizing metal ions into a microfiber nonwoven fabric (MNWF) made through radiation-induced graft polymerization. The suitable metal complex was selected according to the data in “Cambridge Crystallographic Data Center (CCDC)”. 4-picolylamine (4-AMP), designated as a ligand through the metal complex data of CCDC, was introduced at an average mole conversion rate of 63%, and copper ions were immobilized at 0.51 mmol/g to the maximum. It was confirmed that degree of grafting (dg) 170% 4-AMP-Cu MNWF, where copper ions are immobilized, can adsorb up to 50% of acetone gas at about 50 ppm, 0.04 mmol/g- 4-AMP-Cu-MNWF, at room temperature and at a ratio of copper ion to adsorbed acetone of 1:10.

## 1. Introduction

As air pollution is of worldwide concern, interest in indoor air quality is increasing in both industry and general households. There are various causes of indoor air pollution, but volatile organic compounds (VOCs) are the biggest problem among them. VOCs have a low boiling point in the liquid or gas phase and are readily present in the atmosphere [1]. Depending on their molecular structure, VOCs include alkanes, alkenes, aromatic hydrocarbons, alcohols, aldehydes, and ketones [1,2,3]. VOCs range from various solvents used in industrial areas to organic gases emitted during production, especially in chemical, petrochemical, pharmaceutical, food processing, pulp and paper, color printing, and painting industries, pharmaceutical factories, insulation, and the automobile industry [2,3,4,5]. Indoor air pollution is a severe problem in modern society, as most of our daily life takes place indoors. In particular, VOCs, which are indoor air pollutants, are affected by the environment, such as closed places and humidity. The concentration can be higher indoors than outdoors, resulting in dangerous conditions [6,7,8]. The most well-known term associated with indoor air quality pollution is sick building syndrome (SBS). SBS was first defined by the WHO in 1982 and described a set of symptoms that are common indoors [9]. The VOCs considered as the leading causes of SBS are formaldehyde, acetaldehyde, benzene, ethylbenzene, limonene, p-dichlorobenzene, styrene toluene, xylene, and acetone. These are mainly generated from building materials, paint, ink for printing, or smoke from cooking [10]. Since it is difficult to be completely free from VOCs in our daily life and in industry, we aimed at the detection and adsorption of VOCs.

First, acetone was selected from the list of VOCs as a target molecule. Acetone, one of the simplest ketones, affects human health when the concentration exceeds 173 ppm [11]. Previously, acetone was treated as a VOC; nowadays, many countries have excluded acetone from the VOC list. However, this does not mean it is without risk. Acetone is mainly used in industrial sites and laboratories, and the general public can also purchase it in a diluted state. Short-term exposure to acetone irritates the central nervous system, aggravates the nose, throat, lungs, and eyes, and causes headaches and dizziness. It may also produce vomiting symptoms. If very high acetone levels are inhaled or exposed, one may lose consciousness and experience skin irritation [12]. Therefore, it is necessary to control acetone concentration in the living environment or workplace [13]. In addition, acetone can function as a biomarker for diseases through the amount contained in the breath when a person exhales. It has been found that when the amount of acetone in the expiratory volume differs from that of ordinary people, it can indicate the risk of lung cancer, diabetes, dietary fat loss, congestive heart failure, and cerebral attack [14,15,16,17]. Although research to develop a very-high-sensitivity sensor to function as a biomarker of a disease is being actively conducted, the effective high-sensitivity acetone detection sensor proposed in modern study is bulky and has several significant technical problems, such as limited operation at high temperatures. Therefore, there is a need for a sensor that can detect acetone with high sensitivity at room temperature and at a low cost [18,19,20,21].

As a recent trend in gas sensing research, research on VOC sensors using metal oxides has proven fruitful. Gas sensing using metal oxides also has some disadvantages. It is challenging to use only a single metal oxide material, and when used in an un-doped state, there is increased resistance [22,23,24]. Inspired by the latest trends in gas sensing using metal oxides, we devised a method to develop materials that can be used for sensing using metal ions. To create a material capable of VOC adsorption and detection, it was decided to use a radiation-induced graft polymerization technique rather than a metal oxide-based material. Radiation-induced graft polymerization is a method that utilizes radicals generated when a base substrate is irradiated with radiation, plasma, light, and chemicals in a way so as to introduce various functionalities. The polymer chains extend from the base polymer surface, called a polymer brush. The polymer brush can react with the functional group, and its performance can be improved by adjusting its length, density, etc. [25]. The base polymers that are primarily used include polyethylene, polypropylene, polytetrafluoroethylene, nylon 6, etc. [25], and the shapes of the base polymers applied here vary, including porous hollow fiber membranes, nonwoven fabrics, films, and nanotubes. They have been used in the past for experiments with adsorbents or detection sensors [26,27,28,29,30,31,32,33]. In particular, graft polymerization was performed using vinyl monomer glycidyl methacrylate (GMA), containing an epoxy group as a monomer [26]. Therefore, radiation-induced graft polymerization using GMA was considered in this experiment. For the polymer base material, we selected microfiber nonwoven fabric (MNWF) made of polypropylene. In general, the advantage of the nonwoven fabric is that it has large pores compared to other substrates, so it is not easily clogged and has strong durability [28]. As a disadvantage, since the surface area of the functional group bonded to the nonwoven fabric in contact with the adsorption target is smaller than that of other substrates, the impression is that it is disadvantageous for adsorption or sensing. However, in modern experiments, there have been various attempts, such as shape conversion or coating, to compensate for these characteristics of the nonwoven fabric. This experiment used a nonwoven fabric, in which fibers are woven at microfiber levels.

When the radiation-induced graft polymerization method was applied, it was expected that a high-density polymer brush could be introduced into the microfiber nonwoven fabric and that metal ions would be immobilized on the functional group present in the polymer brush. Additionally, unlike general metal ion immobilization experiments, our experiment needed an appropriate bonding level between the functional group and the metal ion. If metal ions are too strongly bonded to the functional group, coordination sites capable of bonding to acetone gas are not expected to remain. The immobilization of metal ions is only part of imparting functionality to microfiber nonwoven fabric, and the final goal is to attempt the adsorption of acetone gas, which is one of the VOCs. Figure 1 shows an expected image of a material that could try to immobilize a metal ion with an appropriate bonding level using a functional group introduced in a polymer brush in the final step and adsorb acetone gas on an immobilized metal ion.

## 2. Materials and Methods

### 2.1. Materials

The materials used comprised 4-picolylamine (98%, Tokyo chemical industry Co., Ltd., Tokyo, Japan, 97.5%, Alfa Aesar Chemical Co., ltd., Haverhill, USA), disodium iminodiacetate hydrate (98%, Tokyo Chemical Industry Co., Ltd., Tokyo, Japan), sulfuric acid (0.5 mol/L, KANTO CHEMICAL Co., INC., Tokyo, Japan), hydrochloric acid (1 mol/L, Tokyo Chemical Industry Co., Ltd., Tokyo, Japan), GASTEC GV-100, GASTEC No.151L, polypropylene microfiber nonwoven fabric (dg 30~520%, ENEOS Corporation, Tokyo, Japan), acetone (99.5%, Wako Chemical Co. Ltd., Odawara, Japan), ICP ( Model-9820, SHIMADZU, Kyoto, Japan), FE-SEM-EDS (JSM-7800f, Jeol, Tokyo, Japan), gas chromatography (GC-2014, SHIMADZU, Kyoto, Japan).

### 2.2. Methods

#### 2.2.1. Selection and Introduction of a Functional Group on GMA MNWF

GMA MNWF was provided by ENEOS corporation; the concentration of GMA used was 6.5~10%, and the irradiation dose was 10~20 kGy. Each prepared GMA MNWF was classified and used using only to the degree of the grafting value. The dg used in the experiment was 30~523%. The degree of graft is given by Equation (1):(1)Degree of graft (dg)=100(W1−W0)/W0 [%] 
where W0 is the mass of the polymer that has not been treated, and W1 is the mass of the polymer after reacting with the GMA monomer [25].

First, metal ions and ligands thought to be capable of binding acetone to the provided GMA MNWF were searched using CCDC. Although there are many metal ions theoretically presumed to be capable of binding to acetone, the list of substances that are likely to bind acetone in the state in which the metal ions discovered through CCDC are bound to ligands include copper, nickel, manganese, zirconium, lanthanum, etc. Since many studies on the immobilization of copper ions to polymer brushes have been conducted previously, it was judged that using copper ions for immobilization was preferable for comparison of the amount of immobilization.

After selecting copper ions as the immobilization target, we identified a copper-immobilized ligand binding to acetone. Since some of the searched ligands can exist only theoretically, substances with similar structures were selected by comparing them with practically existing substances, and 4-picolylamine (4-AMP) was finally selected. The introduction process of 4-AMP chosen as a functional group is shown in Figure 2. When MNWF is irradiated with an electron beam, radicals are generated on the surface. When it is polymerized with glycidyl methacrylate (GMA), a monomer in a state in which radicals are generated, a polymer brush is developed on the surface of MNWF, which is expressed as GMA MNWF. By reacting GMA MNWF with a 4-AMP solution of 1 M concentration at 353 K for 24 h [34], 4-AMP MNWF can finally be obtained. The amount of 4-AMP introduced into GMA MNWF was calculated through the molar conversion rate. The molar conversion rate is given by Equation (2): (2)Molar conversion rate=100[(W2−W1)/108.1]/[( W1−W0)/142.2] [%] 
where W0 is the mass of the polymer that has not been treated, W1 is the mass of the polymer after reacting with the GMA monomer, and W2 is the mass of the polymer after the introduction of the functional group by the addition of a 4-AMP to the epoxy group. The value of 108.1 and 142.2 are the molecular masses of 4-AMP and GMA, respectively [25].

#### 2.2.2. Copper Ion Immobilization on MNWF

The experimental method of the study with high copper ion immobilization efficiency was consulted. The referenced research used an iminodiacetate (IDA) group to immobilize copper ions by reacting 0.01 M CuSO4 aqueous solution at 303 K [35]. To check the maximum copper ion yield of 4-AMP MNWF, the reaction time was varied from 2 h to 24 h, and the results were confirmed and compared. Table 1 shows the 4-AMP and copper ion immobilization conditions in the GMA-grafted MNWF.

#### 2.2.3. Acetone Gas Adsorption

Tedlar bags of 50 L and 5 L were used for the acetone gas adsorption experiment. First, nitrogen gas-based acetone gas was prepared by filling a 50 L Tedlar bag with nitrogen and injecting a certain amount of an acetone solution with a purity of 99.5%. The acetone gas used in the experiment was used after giving it a stabilization time of at least 12 h after manufacturing. The concentration of the completed acetone gas was measured through the gastec detection tube, and the final concentration of the completed acetone gas was about 40 to 60 ppm. The experiment was conducted at an average concentration of 50 ppm. MNWF (GMA MNWF, 4-AMP MNWF, 4-AMP-Cu MNWF) for each stage was put into a 5 L Tedlar bag, and a vacuum state was created; then, 4 L manufactured acetone gas was inserted into the 5 L Tedlar bag to observe the change in concentration. The acetone gas prepared at a concentration of 50 ppm had a concentration of 0.005%, and it was changed into mg/L unit using this concentration. Then the number of moles of acetone gas used in the experiment was obtained. Calculating the number of moles was converted using by Equation (3):(3)$$Concentration\;of\;acetone\;gas\;\left[mg/L\right]=0.005\left(\frac{M}{22.4}\right)\left(\frac{273}{273+T}\right)\left(\frac{P}{1013}\right)\;$$
where 0.005 is a value converted from a unit of 50 ppm to a % unit, M is the molecular weight of acetone, 58.1, 22.4 are the volume of 1 mole of a molecule at 1 atm, 273 is a Kelvin value, T is the temperature used in the experiment, P is 1—the atmospheric pressure at the measurement point, and 1013 is the hPa value at 1 atmosphere.

## 3. Results and Discussion

### 3.1. Selection of Functional Group and Metal Ions

The functional groups to be introduced into the GMA MNWF were selected through a search in the CCDC database. First, a metal complex capable of binding to acetone was identified in the CCDC database. Among the search results, ligands that can be introduced into the GMA MNWF were selected; the referenced data from the CCDC database are shown in Figure 3.

We classified ligands capable of bonding to copper ions, of which ligands applicable to experiments and likely to exhibit high metal immobilization properties were selected. As a result of reviewing the price and purchase route of several chemicals expected to have an appropriate level of binding capacity with copper ions by referring to the reference related to CCDC Refcode HECYUJ [36], it was decided to use 4-AMP for pyridine-based chemicals. Since 4-AMP also possesses the characteristics of the amine group, it seemed that it would facilitate the immobilization of metal ions. In addition, it was expected to show high introduction capability in the GMA with an epoxy group-grafted MNWF because it has an amino group [37]. The method of introducing functional groups was determined based on research data on the bonding of metal ions with 4-AMP or 2-picolylamine, similar to 4-AMP’s molecular structure [33,34,38,39]. The process of selecting 4-AMP using the results of the CCDC database is shown in Figure 4.

### 3.2. Introduction of Functional Group, 4-AMP, on GMA MNWF

A concentration of 1 M of 4-AMP was introduced into GMA MNWF with each degree of grafting, and the introduction characteristics of 4-AMP dependent on the degree of grafting were confirmed. These results are shown in Figure 5. In the case of 4-AMP MNWF, a tendency toward the high introduction of 4-AMP was confirmed as the degree of grafting increased. The highest introduced 4-AMP density on MNWF was about 25.7 mmol/g at dg 523%. This seems to be because, as the degree of grafting increases, GMA content increases and reacts with more 4-AMP.

The color of the MNWF-incorporating 4-AMP became darker as the degree of grafting increased. A color change was observed in the case of dg less than 200%, and it was impossible to distinguish from the dg over 200% of the MNWF visually. The color change of the MNWF, depending on dg, is shown in Figure 6.

From all results, the molar conversion rate of 4-AMP MNWF from GMA-grafted MNWF was around 63% on average, maintaining a high level without significant change, depending on the degree of grafting. In the radiation-induced graft polymerization experiment using MNWF provided by ENEOS Corporation, the original shape of MNWF is still maintained when the degree of grafting is significantly changed, and the functional group is also introduced while retaining a high molar conversion rate.

### 3.3. Copper Ion Immobilization on 4-AMP MNWF

To confirm the appropriate reaction time between 4-AMP MNWF and the copper ion solution, the change in adsorption amount was observed up to 24 h, including the reaction time of 2 h suggested in the reference paper. The copper ions were adsorbed to 4-AMP MNWF reacted with 1M HCl at a speed of 50 rpm at 293 K for 2 h, eluted, and then diluted 10 times to measure through ICP-OES. The copper concentration of 1M HCl through ICP-OES was converted into mmol units using copper molecular weight and then divided by the 4-AMP-Cu MNWF mass used in the experiment to measure the copper density on MNWF. The change in copper ion immobilization amount over time is shown in Figure 7. In the copper ion immobilization reaction over time, the 4-AMP MNWF reacted for 24 h can immobilize much more. As a result, Figure 7 does not confirm whether the amount of copper ion adsorption is in a saturated state; there is a possibility that a higher adsorption amount may appear when it reacts with copper ions for more than 24 h. However, this study is not aimed at observing the maximum adsorption amount of copper ions. In addition, considering that the production process of each functional nonwoven fabric is required in units of 24 h, the adsorption time of copper ions was set to a maximum of 24 h for efficient time distribution.

Figure 8 shows the results of copper ion density on 4-AMP MNWF, depending on 4-AMP density on GMA MNWF. Since 4-AMP was introduced in a large amount while maintaining a high mole conversion rate in an experiment introducing 4-AMP on GMA MNWF, copper ion immobilization was expected to occur more in high dg 4-AMP MNWF. As a result of comparing and calculating the number of moles of copper ions depending on the number of moles of 4-AMP, the maximum copper ion immobilization amount was confirmed when the density of the amount of 4-AMP was about 7–13 mmol/g. The highest copper ion immobilization result was observed in 4-AMP-Cu MNWF containing dg 240%. At this time, 4-AMP was 12.75 mmol/g, and copper ions were immobilized at 0.51 mmol/g. According to the references, the copper adsorption amount was 0.42–0.44 mmol/g for IDA-Cu [34] and 0.55 mmol/g for EDA-Cu [40], and recent studies have continued to attempt to increase the adsorption amount of copper ions by adding several processes [41]. The maximum adsorption amount observed in our experimental results is considered to be equivalent to that of adsorbents using other functional groups.

In the case of 4-AMP-Cu MNWF (dg 160–200%), the copper ion immobilization amount was 0.30–0.44 mmol/g, and 4-AMP was 7–10 mmol/g. At this time, the ratio of copper ions to 4-AMP was 1:20–24. The total adsorption amount of copper was lower than dg 240% 4-AMP-Cu MNWF, but the ratio of copper to 4-AMP decreased, which is thought to have been introduced with the appropriate binding force at the desired level. A decreasing trend was observed after showing the maximum amount of copper ion immobilization in dg 240% 4-AMP MNWF. From the analysis of the results, when each 4-AMP is 21, 25 mmol/g, the immobilized amount of copper ions is 0.44, 0.47 mmol/g, and the ratio of copper ion to 4-AMP is 1:44 and 1:56. The following assumptions can be made as to the reason for the decrease in the efficiency of the immobilization of copper ions. There are limited coordination sites on copper ions that can immobilize at functional groups and an acetone molecule. As the density of 4-AMP increases, the copper ion coordination site is occupied by 4-AMP molecules, which seems to be why the amount of copper ion immobilization does not increase in high dg 4-AMP MNWF. Based on the copper ion density results depending on 4-AMP density, the copper ion immobilization efficiency seems to be the most effective when the ratio is about 1:20–25.

Figure 9 shows SEM images of GMA MNWF, 4-AMP MNWF, and 4-AMP-Cu MNWF. SEM images of GMA MNWF, 4-AMP MNWF, and 4-AMP-Cu MNWF were similar in shape, making it difficult to distinguish each stage with each image alone.

### 3.4. Acetone Gas Adsorption and Desorption

MNWF introduced with GMA, 4-AMP, and 4-AMP-Cu was put in a 5 L Tedlar bag, and 4 L acetone gas was injected to measure changes over time at room temperature. In the preliminary experiment stage, the acetone gas concentration inside the Tedlar bag tended to decrease to 1 h from the start of the reaction, and since it was confirmed that the acetone gas concentration inside the Tedlar bag increased again after 1 h, the maximum time for the adsorption reaction was set to 1 h. In the case of the change in adsorption amount, 2 mL of acetone gas inside the Tedlar bag was extracted for predetermined times at intervals of 10, 20, 40, and 60 min, and the concentration change was measured by gas chromatography. In the desorption process, the maximum observation time was set equal to 1 h, and the amount of change was checked every 30 min.

#### 3.4.1. Each Functional MNWF’s Acetone Gas Adsorption

After putting the prepared acetone gas in a 5 L Tedlar bag, the change in the acetone gas concentration in the absence of a functional MNWF was first checked. The acetone gas concentration was changed by about 1–10% in the state where the functional MNWF was not added. For each functional MNWF, GMA, 4-AMP, and 4-AMP-Cu were used, and the results are shown in Figure 10.

In the case of the GMA MNWF, there was virtually no adsorption capacity considering the variability (10%) when nothing was added. As for 4-AMP MNWF, the maximum adsorption amount was about 20–30% of the total acetone gas concentration, and the adsorption performance was not excellent. 4-AMP-Cu MNWF could adsorb up to 50% of acetone gas, about two times that of 4-AMP MNWF. Even if the correction value was applied, it was about 40%, which showed higher efficiency than 4-AMP MNWF. Each functional MNWF reached the maximum adsorption amount about 10 min after the acetone gas adsorption reaction and then maintained the adsorption level until 1 h after the beginning of the reaction. The above results confirmed that 4-AMP MNWF and 4-AMP-Cu MNWF could adsorb acetone gas at room temperature, and 4-AMP-Cu MNWF has better acetone gas adsorption efficiency.

The change in acetone gas adsorption amount depending on the dg is shown in Figure 11. The data used in Figure 11 were prepared using those with good ratios of copper and 4-AMP among samples in 4-AMP-Cu MNWF of average dg 40, 100, 170, and 240%. The maximum acetone gas adsorption of 4-AMP-Cu MNWF was observed at dg 170% (Cu:4-AMP, 1:20). In the dg 160–200% 4-AMP-Cu MNWF, which showed the highest efficiency in copper ion adsorption, acetone gas also showed the maximum adsorption amount; after that, it decreased. The hypotheses for the adsorption reduction of acetone gas at 4-AMP-Cu MNWF depending on dg change can be considered as follows. The most critical factor in the prepared 4-AMP-Cu MNWF is that acetone is adsorbed to the vacant coordination sites of copper ions immobilized to MNWF. When the dg is too high, too much 4-AMP is introduced, and more 4-AMP is bound to each copper ion, which causes the absence of coordination sites of copper ions. For low dg 4-AMP-Cu MNWF, the copper to 4-AMP ratio sometimes yields similar results to high dg 4-AMP-Cu MNWF. However, since the absolute amount of copper ions immobilized is insufficient, copper ion coordination sites that can bind acetone gas are also lacking. Thus, the 4-AMP-Cu MNWF of too-low dg or too-high dg results are similar to the adsorption amount of 4-AMP MNWF in which copper is not immobilized. Combining the above assumptions and results, when preparing 4-AMP-Cu MNWF for acetone gas adsorption, the appropriate dg of MNWF is 160–200%, and the immobilized copper ion and the 4-AMP ratio is 1:20 or less.

According to Equation (3), the value of mg/L of acetone inside the Tedlar bag was calculated, and the total amount was confirmed. And, it was confirmed that the concentration of 4L of 50 ppm acetone gas injected into the 5 L Tedlar bag was 0.48 mg/L. The dg 170% 4-AMP-Cu MNWF showed the highest acetone gas adsorption, about 0.04 mmol/g- 4-AMP-Cu MNWF, and the ratio of copper ion to adsorbed acetone was 1:10.

Our newly designed 4-AMP-Cu MNWF has yet to have a standard point to compare performance to other acetone gas sensor-based references. A high-sensitivity acetone sensor was operated at a high temperature in general, and the efficiency was confirmed by checking the response according to the temperature in a specific concentration of acetone gas [42,43,44], and when using systems such as bio-filtration, the efficiency is observed through the concentration of the purified gas [45,46]. The focus of this study is to adsorb acetone gas using the empty coordination site of metal ions immobilized on the polymer brush. The resulting 4-AMP-Cu MNWF adsorbed up to 50% of 50 ppm acetone gas at room temperature, which seems valuable because it was able to adsorb 0.04 mmol/g-4-AMP-Cu MNWF.

#### 3.4.2. 4-AMP-Cu MNWF Acetone Gas Desorption

After 1 h of the adsorption experiment, the acetone gas inside the Tedlar bag was removed using a vacuum pump, and a vacuum state was created. After that, the change in acetone gas concentration was observed by filling it with fresh air. The desorption process of 4-AMP-Cu MNWF was measured by gas chromatography by extracting 2 mL of gas at 30 and 60 min after the start of the reaction, respectively. The air used in the experiment had an acetone gas concentration of about 3% (based on 50 ppm). Additionally, the concentration of acetone gas in the Tedlar bag without functional MNWF fluctuated up to 13%. Therefore, in the experimental stage, it was assumed that about 3 to 13% of acetone gas was already present in the Tedlar bag, and the experiment results were confirmed. 4-AMP-Cu MNWF of dg 170% (Cu: 0.37 mmol/g, 1:20), which showed the best results in the adsorption experiment, was used in the desorption experiment. The time-dependent acetone gas desorption change is shown in Figure 12.

In the case of dg 170% 4-AMP-Cu MNWF used in the experiment, it was confirmed that about 43% of gas was desorbed based on a 50 ppm concentration of acetone gas in the desorption process of 1 h. Even excluding the maximum correction value of acetone gas of 13%, which was considered to exist initially, it showed a desorption value of 30%. Since the amount of change in the adsorption and desorption processes was confirmed based on the acetone gas concentration of 50 ppm, the respective results were compared together. Compared to the maximum adsorption amount of acetone gas of 50%, 43/50 (using the correction value, 30/40), or about 86% (using the correction value, 75%) of the adsorption amount was desorbed. The desorption experiment performed at room temperature confirmed that the acetone gas adsorbed to the 4-AMP-Cu MNWF was easily desorbed without a separate separation process. This confirmed that the 4-AMP-Cu MNWF and the acetone gas had an appropriate binding level. It is judged that such an appropriate binding level can be advantageous for recycling because additional costs and processes are not required in adsorption and desorption.

#### 3.4.3. Whether 4-AMP-Cu MNWF Can Be Reused

Reuse testing was performed using dg 170% 4-AMP-Cu MNWF, which exhibited the highest acetone gas adsorption rate. In each reuse experiment, adsorption and desorption were performed as one process, and this process was repeated three times. In the secondary adsorption, the adsorption rate compared to the total acetone gas concentration decreased by about 7% compared to the primary adsorption (50% -> 43%). Similarly, in the third experiment, the adsorption amount decreased by about 6% compared to the adsorption in the second experiment (43% -> 37%). The adsorption amount of acetone gas for each cycle is shown in Figure 13.

As the cycle progressed, the adsorption efficiency tended to decrease slightly. The decrease in adsorption efficiency appears to be due to strong binding of some acetone gases to 4-AMP-Cu MNWF, resulting in limited desorption.

## 4. Conclusions

This study started from the possibility that VOCs could be efficiently removed and adsorbed by introducing metal ions into microfiber nonwoven fabric, focusing on the fact that metal oxide-based sensors can remove and adsorb VOCs. To be repeatedly used at room temperature, MNWF of polypropylene with good physical durability was used as a base material, and a functional group estimated to be capable of adsorbing VOCs in a state of immobilized metal ions was selected through the Cambridge Crystal Data Center (CCDC).

In the case of the functional group 4-AMP, selected according to CCDC metal complex data, a large amount was introduced while maintaining a high mole conversion rate (avg 63%) even at high dg MNWF. When the ratio of copper ions to 4-AMP was 1:25 at dg 240% 4-AMP-Cu MNWF, the maximum immobilized amount of copper ions was 0.51 mmol/g.

Both adsorption and desorption experiments of acetone gas and volatile organic compounds (VOCs) were conducted at room temperature, and up to 50% of gas could be adsorbed at a concentration of about 50 ppm of acetone gas. The maximum adsorption amount of acetone gas was observed when the dg of 4-AMP-Cu MNWF was 160–200% and the ratio of copper ions to 4-AMP was 1:20. When the maximum amount of acetone gas adsorption was observed, 4-AMP-Cu MNWF could adsorb about 0.04 mmol/g- 4-AMP-Cu-MNWF at room temperature, and the ratio of copper ions to adsorbed acetone was 1:10. In addition, most adsorption acetone gases are desorbed by about 86% of the adsorption amount just by being exposed to the air, without a separate process.

4-AMP-Cu MNWF is an initial stage of technology development capable of adsorbing and desorbing VOCs using a ligand containing metal ions. However, for 4-AMP-Cu MNWF, more research is needed because it was confirmed that volatile organic compounds can be repeatedly adsorbed and desorbed at room temperature. Optimal 4-AMP introduction conditions must be identified, and studies on selective adsorption with other metal ions or VOCs are also needed. In the near future, we will study selectively adsorbed formaldehyde, one of the leading causes of SBS, by adsorbing other metal ions on MNWF based on 4-AMP functional groups. We expect that the adsorption properties of 4-AMP MNWF with other metal ions and the characteristics of adsorption of other VOCs will be identified through formaldehyde gas adsorption experiments to be carried out in the next step.

## Figures and Tables

**Figure 1 sensors-22-00091-f001:**
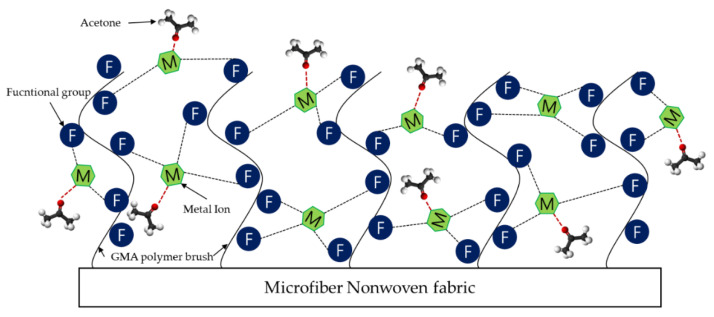
The design concept of a new material in which acetone molecules coordinate to metal ions immobilized on functional microfiber nonwoven fabric.

**Figure 2 sensors-22-00091-f002:**
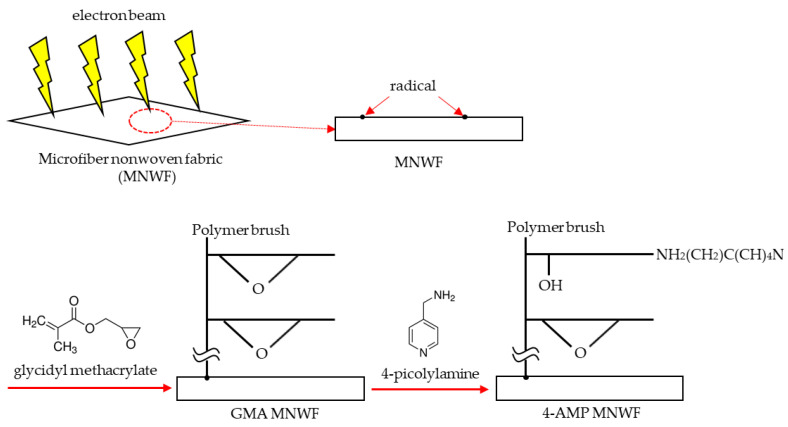
Introduction of 4-AMF into GMA MNWF.

**Figure 3 sensors-22-00091-f003:**
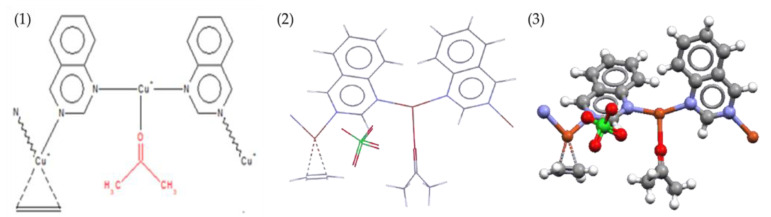
Image of referenced CCDC data (Refcode: HECYUJ, (**1**): 2D image, (**2**): 3D image, (**3**): 3D image).

**Figure 4 sensors-22-00091-f004:**
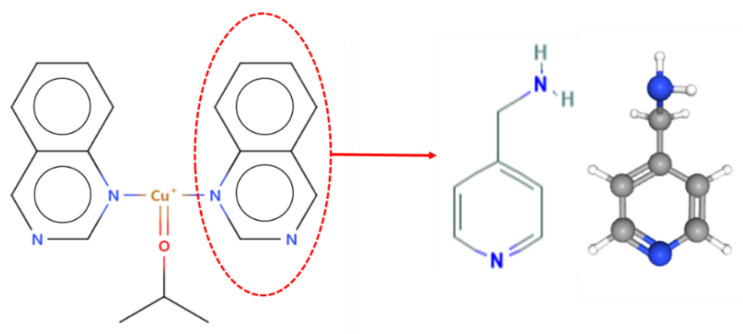
4-AMP’s selection process (CCDC Refcode: HECYUJ).

**Figure 5 sensors-22-00091-f005:**
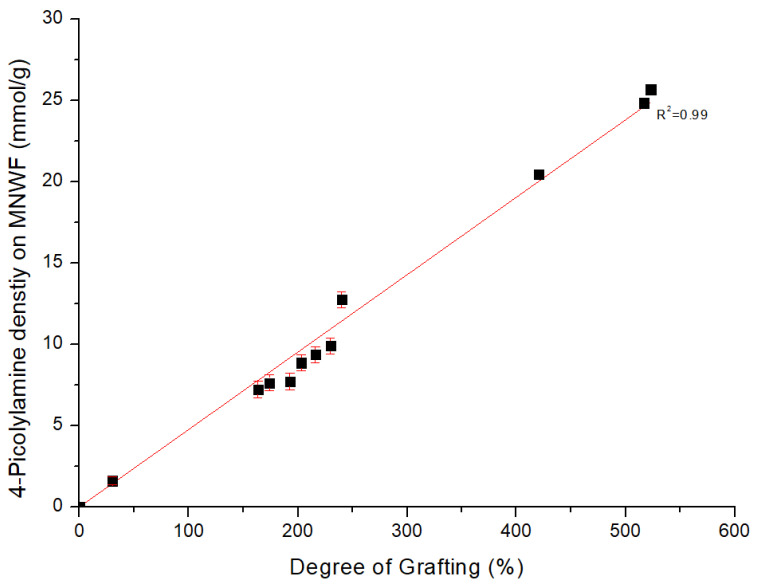
Introduction of 4-AMP on GMA MNWF.

**Figure 6 sensors-22-00091-f006:**
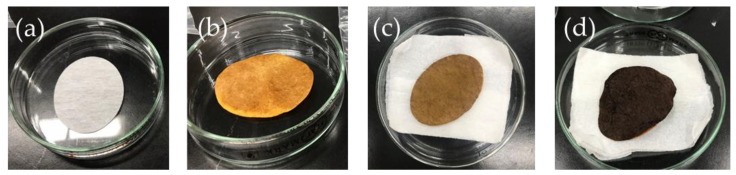
Color change of MNWF depending on the degree of grafting (**a**): dg 100% GMA MNWF, (**b**): dg 100% 4-AMP MNWF, (**c**): dg 170% 4-AMP MNWF, (**d**): dg 200% 4-AMP MNWF.

**Figure 7 sensors-22-00091-f007:**
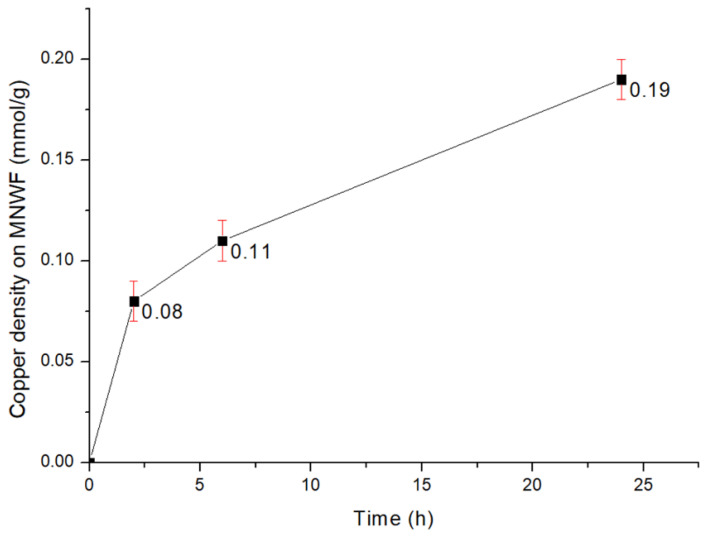
Change in copper ion immobilization amount over time (dg 30% 4-AMP MNWF).

**Figure 8 sensors-22-00091-f008:**
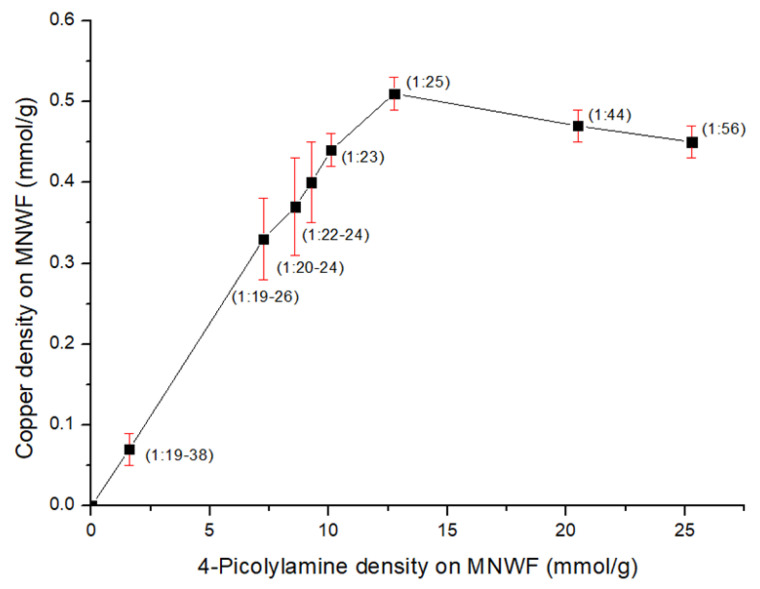
Dependence of copper ion density on 4-AMP density (the content in parentheses is the ratio of Cu mole:4-AMP mole).

**Figure 9 sensors-22-00091-f009:**
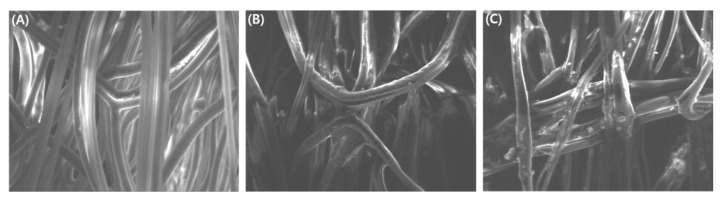
SEM image of each functional MNWF. (**A**): GMA MNWF, (**B**): 4-AMP MNWF, (**C**): 4-AMP-Cu MNWF. Each of MNWF’s dg is 170%.

**Figure 10 sensors-22-00091-f010:**
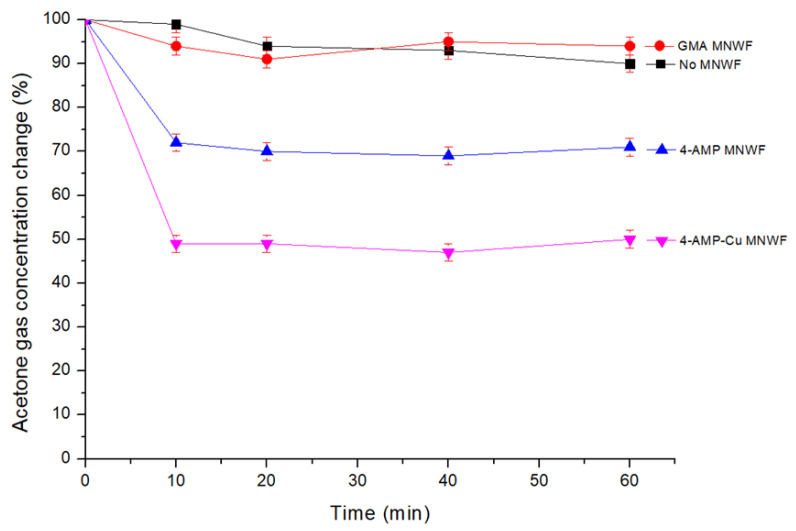
Acetone gas adsorption amount of each MNWF (about 50 ppm, dg 170% MNWF).

**Figure 11 sensors-22-00091-f011:**
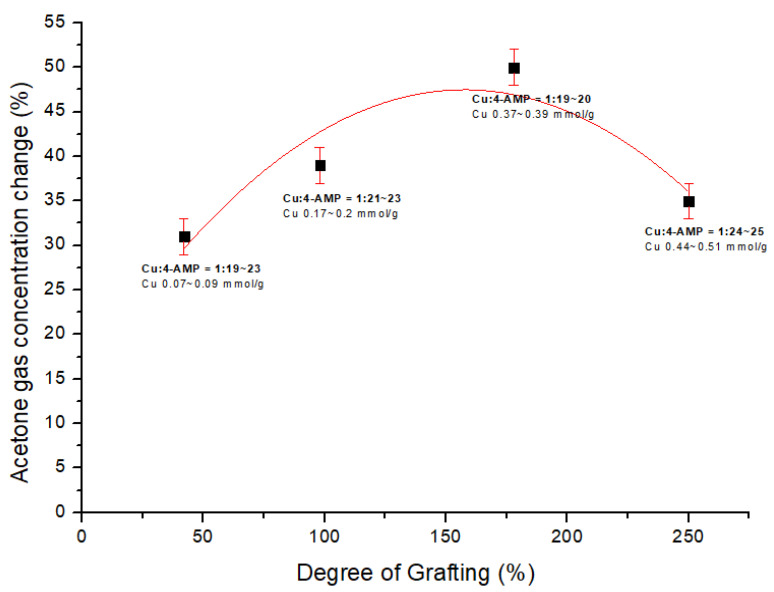
Dependence of acetone gas adsorption on the degree of grafting (about 50 ppm).

**Figure 12 sensors-22-00091-f012:**
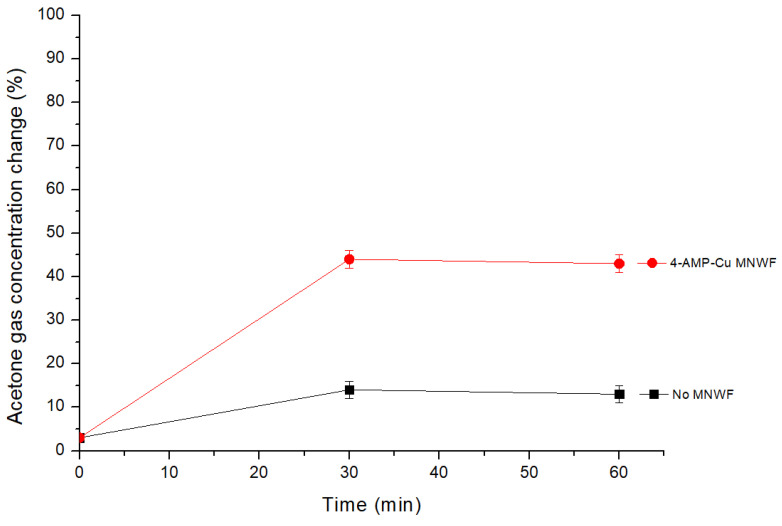
Acetone gas desorption amount of 4-AMP-Cu MNWF (about 50 ppm).

**Figure 13 sensors-22-00091-f013:**
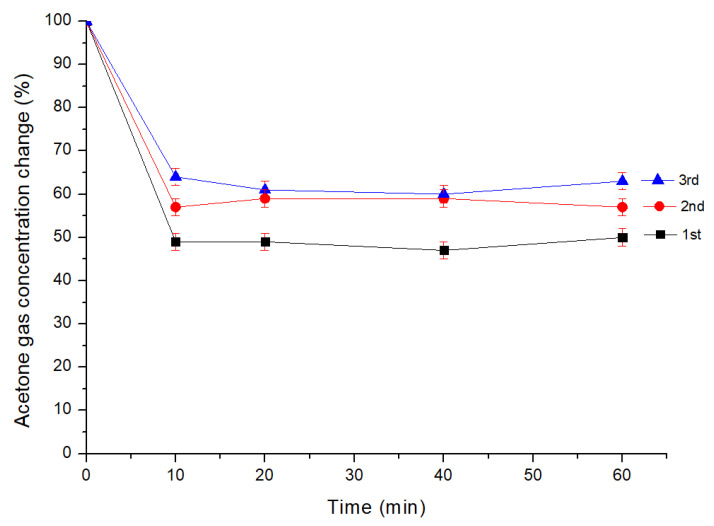
Acetone gas adsorption amount for reuse tests (dg 170% 4-AMP-Cu, 50 ppm).

**Table 1 sensors-22-00091-t001:** Introduction of 4-AMP and copper ion immobilization in the GMA MNWF.

**Grafting**	
Degree of grafting	30~523%
**Introduction of 4-picolylamine**	
Concentration of 4-picolylamine	1 M
pH	10.5–11.5
Reaction temp	353 K
Reaction time	24 h
**Metal chelation**	
Concentration of CuSO4	0.01 M
Temp	303 K
Reaction time	24 h

## Data Availability

Not applicable.

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
