# Peer review of "Functional Microfiber Nonwoven Fabric with Copper Ion-Immobilized Polymer Brush for Detection and Adsorption of Acetone Gas"

_sensors, 2021, doi:10.3390/s22010091_

Round 1
Reviewer 1 Report
In the manuscript entitled “New materials design for detection or adsorption of acetone gas”, authors developed a new material capable of detecting or adsorbing VOCs in a state in which the metal complex between functional groups and metal ions are immobilized on microfiber nonwoven fabric (MNWF) using a radiation induced graft polymerization method. After going through the complete manuscript, herein summarize the points which may improve the quality of the manuscript and the readability to the reader. The manuscript needs a major revision before considering for the acceptation in the journal of “Sensors”.
However, some points are mentioned as follows for the betterment of the manuscript.
- The title of the manuscript should be concise, informative, effective and create an impact to the reader. Kindly rephrase it if possible.
- The abstract should be more informative rather than descriptive, please mention the specific findings about the study so that the reader would easily understand the rationale of the study.
- The Introduction section is well written but the authors authors need to improve some sections, also include all the recent data and cite it properly with the literature relevant with the present research which improves the credibility of the paper.
- In the result section, some sentences and phrases are not clear, please correct them.
- In Figure.3, the authors need to rewrite the labeling properly as it is not clearly visible. Also label all the figures properly as per the journal guidelines.
- In result and discussion section, Page – 13, Line no – 280, the text should be justified properly.
- In conclusion section, the authors should compare their findings with the already published literature work and conclude in a precise and effective way.
- In case of Bibliography/Reference section, the reference no-1,3,6,7, 10, 11, 12, 15, 28, the volume is missing. Please refer the journal guidelines and recheck all the references mentioned in the manuscript.
- The article requires a thorough check the grammatical, English and typological errors.
Author Response
First of all, we would like to thank the reviewer for taking the time to direct me to the correction.
Including the nine problems pointed out, we have revised the overall contents of the thesis.
1. The title of the manuscript should be concise, informative, effective and create an impact to the reader. Kindly rephrase it if possible.
Answer: As you suggested, we have changed our title more clearly.
2. The abstract should be more informative rather than descriptive, please mention the specific findings about the study so that the reader would easily understand the rationale of the study.
Answer: As you suggested, we wrote down the specific research results in the abstract part to make it easier to understand.
3. The Introduction section is well written but the authors authors need to improve some sections, also include all the recent data and cite it properly with the literature relevant with the present research which improves the credibility of the paper.
Answer: The introduction sentences were generally refurbished, and the reason for the need for acetone detection in human breathing was added.
4. In the result section, some sentences and phrases are not clear, please correct them.
Answer: The English grammar of the entire thesis was corrected, and the formula used in the result column was added to the Method. In addition, the contents of the results and discussion column have also been added and corrected.
5. In Figure.3, the authors need to rewrite the labeling properly as it is not clearly visible. Also label all the figures properly as per the journal guidelines.
Answer: The problem in the previous paper's figure 3 was confirmed and deleted. Instead, in the main text, the content of Figure 3 was introduced in detail in writing.
6. In result and discussion section, Page – 13, Line no – 280, the text should be justified properly.
Answer: Since the same error was confirmed in several places and the part you pointed out, we corrected and reviewed them and sorted them.
7. In conclusion section, the authors should compare their findings with the already published literature work and conclude in a precise and effective way.
Answer: As suggested, we tried to compare the results by checking other papers for the detection or adsorption of acetone gas. However, the reference papers are different from our sample in the sample manufacturing method or the acetone gas adsorption experiment. Therefore, finding a common reference point for comparing efficiencies wasn't easy. We consider this research to be an experiment to develop a new type of adsorbent that can adsorb acetone gas at room temperature using a polymer brush in which metal ions are immobilized. In this study, a new kind of adsorbent that can adsorb up to 50% of 50ppm concentration of acetone gas at room temperature was created using a polymer brush with metal ions immobilized thereon. Although it was not easy to objectively compare the efficiency with other sensors, it confirmed the possibility of developing a new type of adsorbent, and we think it is a meaningful study with ample room for future development.
8. In case of Bibliography/Reference section, the reference no- 1,3,6,7, 10, 11, 12, 15, 28, the volume is missing. Please refer the journal guidelines and recheck all the references mentioned in the manuscript.
Answer: As you said, we checked the errors in the references section. Therefore, we read the journal guidelines again and modified them accordingly.
9. The article requires a thorough check the grammatical, English and typological errors.
Answer: As you said, we found many grammar errors in our article. Therefore, the grammar was corrected using the English grammar correction service provided by MDPI.
We have tried our best to respond as best as possible to the reviewer, who kindly informed us of the corrections. Above all, thank you for teaching us so much for our development.
Reviewer 2 Report
I have found the manuscript of real interest pointed towards a possible application in evidencing the presence of acetone in various environments.
At the same, before being accepted for publication, I have added more remarks on the annotated manuscript to which I would recommend authors to be included in the revised version.

Author Response
First of all, we would like to thank the reviewer for taking the time to direct me to the correction.
Including the problems pointed out, we have revised the overall contents of the thesis.
We will summarize our problems identified through the comments you left in the PDF file and guide you in answers to each.
1. Add ORCID ID
Answer: I have registered a new ORCID ID. The ORCID ID is 0000-0002-8469-9386.
2. Put number on abstract and how efficiently
Answer: As you suggested, we wrote down the specific research results in the abstract part to make it easier to understand.
3. To 2-2-1, Put References more
Answer: GMA MNWF used in 2-2-1 was provided by ENEOS Corporation and is the same as the method generally used for radiation-induced graft polymerization based on GMA monomer. In addition, references of the introduced 4-AMP introduction conditions found through CCDC were inserted.
4. Explain Fig.2
Answer: The figure has been modified to make it easier to understand and the details have been newly described.
5. How to calculate molar conversion rate?
Answer: All equations for calculating the numerical values used in the results are inserted in the Method section.
6. explain how you measured it with ICP-OEC.
Answer: A description of the method for measuring copper ion concentration using ICP-OES is inserted in the text.
7. Please explain how you measured copper density on MNWF.
Answer: The copper adsorption concentration of 4-AMP-Cu MNWF was confirmed through the copper ion elution amount confirmed through ICP-OES, and the mmol unit was obtained by dividing by the copper molecular weight. The obtained copper value was divided by the mass of 4-AMP-Cu MNWF to complete the unit of mmol/g. This content has been added in addition to the text.
8. Please put the scale seat in the SEM image. Please add contents more
Answer: Scale bars have been inserted into the SEM image. Only the SEM image confirmed this time could not confirm the characteristic change of the surface of the nonwoven fabric at each stage. However, as shown in Figure 6, as dg increased, a distinct color change was observed when 4-AMP was introduced. In our following paper, we plan to confirm the adsorption properties of formaldehyde using 4-AMP and metal ions. XRD or FT-IR analysis will also be performed at this stage, so it is expected that more detailed physical properties will be confirmed.
9. Please put an error value in all graph data.
Answer: All data involved in graph creation were rechecked, and an error bar was inserted.
10. everywhere, the acetone gas concentration seems to obey an saturation exponential, with different saturation time. please comment this finding.
Answer: This is our first attempt at gas adsorption experiments. Therefore, in a preliminary experiment, 4-AMP-Cu MNWF was reacted with acetone gas at a concentration of 50 ppm for 2 hours to observe the progress. As a result of the initial experiment, acetone showed the maximum adsorption after 10 minutes of reaction start and maintained the level until 60 minutes. After 60 minutes, the concentration of acetone gas inside the Tedlar bag started to rise again, and it was confirmed that the concentration of acetone gas when 120 minutes had elapsed from the start of the reaction approached the concentration of 50 ppm again. In this experiment, saturation time and saturation index were not considered. Based on the results of preliminary experiments, research was carried out with meaning in the possibility of developing a new type of adsorbent. When dealing with formaldehyde, which was selected as the adsorption target for the following paper, we will try to refer to the reviewers' opinions to be reflected in the results and discussion.
We have tried our best to respond as best as possible to the reviewer, who kindly informed us of the corrections. Above all, thank you for teaching us so much for our development.

Reviewer 3 Report
In this paper,the author have devised a way to develop materials that can be used for sensing using metal ions,it was decided to use a radiation-induced graft polymerization technique rather than a metal oxides based material. Some severe issues should be addressed before considering acceptance.
- On page 4, line 111, it is suggested that the author add a detailed explanation about Figure 2 to help readers better understand how to introduce functional groups into GMA MNWF.
- In the introduction, the author should consider the role of functionality of acetone in breathing analysis as a biomarker so as to give a comprehensive background and concept introduction. Some relative papers are listed as follow: Adv. Mater., 2021, 33, 2101262.; Nano Energy, 2020, 74, 104941;
- As shown in Figure 8, after 24 hours of reaction, the immobilization rate of copper ion does not slow down, and the copper density should continue to increase if the reaction continues. Has the author tried to increase the reaction time to obtain a higher copper density?
- Some grammar errors and typos appear in the current manuscript. It is suggested to polish the English writing very carefully during the revision.
- It is highly recommended to draw a clear scale bar in Fig. 10 rather than using the original ones in SEM images.
- How about the long-term stability of the prepared device?
- The test for selectivity should be added.
- Please delete the serial number in Fig. 3 and replace them with subcaptions.
- Characterization such as XRD and FTIR should be supplemented to indicate the crystal structure and chemical composition of the sensitive materials.
Author Response
First of all, we would like to thank the reviewer for taking the time to direct me to the correction.
Including the nine problems pointed out, we have revised the overall contents of the thesis.
1. On page 4, line 111, it is suggested that the author add a detailed explanation about Figure 2 to help readers better understand how to introduce functional groups into GMA MNWF.
Answer: The figure has been modified to make it easier to understand and the details have been newly described.
2. In the introduction, the author should consider the role of functionality of acetone in breathing analysis as a biomarker so as to give a comprehensive background and concept introduction. Some relative papers are listed as follow: Adv. Mater., 2021, 33, 2101262.; Nano Energy, 2020, 74, 104941;
Answer: The reason why the detection of acetone gas is necessary for relation to human respiration is added to the introduction.
3. As shown in Figure 8, after 24 hours of reaction, the immobilization rate of copper ion does not slow down, and the copper density should continue to increase if the reaction continues. Has the author tried to increase the reaction time to obtain a higher copper density?
Answer: In the case of the reference paper of the copper introduction experiment, a reaction was conducted with a copper solution for 2 hours. In our case, we checked from 2 hours to up to 24 hours to observe the change in copper adsorption, and as a result, we know that copper may be more adsorbable. In this study, most of the manufacturing processes for each MNWF took up to 24 hours, and our purpose was not to check the maximum copper adsorption, so the adsorption time of copper ions was set up to 24 hours for the convenience of the manufacturing process. However, since we believe that dealing with the maximum adsorption amount of metal ions is also an essential characteristic of 4-AMP, additional information on the adsorption saturation time of copper ions will be covered in the following study.
4. Some grammar errors and typos appear in the current manuscript. It is suggested to polish the English writing very carefully during the revision.
Answer: Answer: As you said, we found many grammar errors in our article. Therefore, the grammar was corrected using the English grammar correction service provided by MDPI.
5. It is highly recommended to draw a clear scale bar in Fig. 10 rather than using the original ones in SEM images.
Answer: Scale bars have been inserted into the SEM image.
6. How about the long-term stability of the prepared device?
Answer: The base material, polypropylene MNWF, is a material with very strong physical durability, and the polymer brush produced by the GMA monomer reaction is also chemically very stable, so we think the long-term stability of the 4-AMP-Cu MNWF is also high.
7. The test for selectivity should be added.
Answer: As a novel type of adsorbent, in this study, we focused and investigated whether the metal ions and functional groups selected through CCDC can adsorb VOCs when immobilized on a polymer brush. In the following paper, we plan to check the selectivity of metal ions and the selectivity of VOCs.
8. Please delete the serial number in Fig. 3 and replace them with subcaptions.
Answer: The problem in the previous paper's figure 3 was confirmed and deleted. Instead, in the main text, the content of Figure 3 was introduced in detail in writing.
9. Characterization such as XRD and FTIR should be supplemented to indicate the crystal structure and chemical composition of the sensitive materials.
Answer: This study was attempted as our new idea, and XRD and FT-IR analysis have not yet been attempted. Because of the results of this study, we plan to try it in the next adsorption experiment for other VOCs such as formaldehyde. Instead, references related to the crystal structure of CCDC's refcod HECYUJ [40], which helped select the 4-AMP functional group, were added.
We have tried our best to respond as best as possible to the reviewer, who kindly informed us of the corrections. Above all, thank you for teaching us so much for our development.

Round 2
Reviewer 2 Report
I have a small suggestion before manuscript be accepted: The refferences does not fulfill the journal template, e.g.
Rezvan Torkaman; Fatemeh Maleki; Mobina Gholami; Meisam Torab-Mostaedi; Mehdi Asadollahzadeh. Assessing the radiation-induced graft polymeric adsorbents with emphasis on heavy metals removing: A systematic, Journal of Water Process Engineering 2021, 44, 102371 [Crossref]
S B Rahardjo et al. Synthesis and Characterization of Tetrakis(2 -amino-3-methylpyridine)copper(II) Sulfate Tetrahydrate, IOP Conference Series: Materials Science and Engineering 2017, 349, 012056 [Crossref]
SHIRVAN, SADIF A; KHAZALI, FEREYDOON; DEZFULI, SARA HAYDARI; BORSALANI, ALI. SYNTHESIS, CRYSTAL STRUCTURE ANALYSIS AND CHARACTERIZATION OF MERCURY(II) COMPLEX CONTAINING 2-(AMINOMETHYL)PYRIDINE AND BROMIDE, Journal of the Chilean Chemical Society 2017, 62, 1, 3350-3353 [Crossref]
It is obvious the authors have "copy and paste" the references from diverse data bases without editing them in accordance with the MDPI template
Author Response
Once again, thank you so much for taking the time to review.
After re-checking the [MDPI Reference List and Citations Style Guide] for your point, we found that the citation style we used was capitalized improperly and the journal name was not italic, so we revised it again to fit the guidelines.
The styles of #35, #36, and #41 have been changed, as you pointed out.
We are very grateful to you for taking the time to review and inform us of our shortcomings in order to advance our research.
